# Relationship between Chinese Herbal Medicine Use and Risk of Sjögren’s Syndrome in Patients with Rheumatoid Arthritis: A Retrospective, Population-Based, Nested Case-Control Study

**DOI:** 10.3390/medicina59040683

**Published:** 2023-03-30

**Authors:** Hou-Hsun Liao, Hanoch Livneh, Miao-Chiu Lin, Ming-Chi Lu, Ning-Sheng Lai, Hung-Rong Yen, Tzung-Yi Tsai

**Affiliations:** 1Graduate Institute of Chinese Medicine, School of Chinese Medicine, College of Chinese Medicine, China Medical University, Taichung 404333, Taiwan; harrisliao@gmail.com; 2Department of Chinese Medicine, Dalin Tzu Chi Hospital, Buddhist Tzu Chi Medical Foundation, Chiayi 62247, Taiwan; 3Department of Nursing, Tzu Chi University of Science and Technology, Hualien 62247, Taiwan; 4Rehabilitation Counseling Program, Portland State University, Portland, OR 97207-0751, USA; livnehh@pdx.edu; 5Department of Nursing, Dalin Tzu Chi Hospital, Buddhist Tzu Chi Medical Foundation, Chiayi 62247, Taiwan; df729376@tzuchi.com.tw; 6School of Medicine, Tzu Chi University, Hualien 97004, Taiwan; dm252940@tzuchi.com.tw; 7Division of Allergy, Immunology and Rheumatology, Dalin Tzu Chi Hospital, Buddhist Tzu Chi Medical Foundation, Chiayi 62247, Taiwan; 8Department of Chinese Medicine, China Medical University Hospital, Taichung 404333, Taiwan; 9Research Center for Traditional Chinese Medicine, Department of Medical Research, China Medical University Hospital, Taichung 404333, Taiwan; 10Chinese Medicine Research Center, China Medical University, Taichung 404333, Taiwan; 11Department of Biotechnology, Asia University, Taichung 41354, Taiwan; 12Department of Environmental and Occupational Health, College of Medicine, National Cheng Kung University, Tainan 70428, Taiwan; 13Department of Medical Research, Dalin Tzu Chi Hospital, Buddhist Tzu Chi Medical Foundation, Chiayi 62247, Taiwan

**Keywords:** rheumatoid arthritis, Sjögren’s syndrome, Chinese herbal medicines, nested case-control study, risk

## Abstract

*Background and Objectives*: Sjögren’s Syndrome (SS) is a common extra-articular feature among subjects with rheumatoid arthritis (RA). While Chinese herbal medicine (CHM) has been used to treat symptoms of RA for many years, few studies have examined its efficacy in guarding against the SS onset. This study aimed to compare risk of SS for RA patients with and without use of CHM. *Materials and Methods*: Data obtained for this nested case-control study were retrieved from Taiwanese nationwide insurance database from 2000–2013. Cases with SS claims were defined and matched to two randomly selected controls without SS from the recruited RA cohorts. Risk of SS in relation to CHM use was estimated by fitting multiple conditional logistic regression. *Results*: Patients aged between 20 and 80 years were included and 916 patients with incident SS were matched to 1832 non-SS controls by age, sex and index year. Among them, 28.1% and 48.4% cases ever received CHM therapy, respectively. After adjusting for baseline characteristics, CHM use was found to be related to a lower risk of SS among them (adjusted odds ratio = 0.40, 95% confidence interval: 0.34–0.47). A dose-dependent, reverse association, was further detected between the cumulative duration of CHM use and SS risk. Those receiving CHM therapy for more than 730 days showed a significantly reduced risk of SS by 83%. *Conclusions*: Findings of this study indicated that the add-on CHM formula, as part of RA care, may be a beneficial treatment for prevention against the incident SS.

## 1. Introduction

Rheumatoid arthritis (RA) is a chronic debilitating inflammatory autoimmune disease that primarily targets joints in the body and other connective tissues. This disorder affects approximately 1% of the population worldwide and is more frequent among women [1]. Just as this chronic disorder has no known cure, estimates indicated that up to 30% of affected patients become permanently work-disabled within the first 2–3 years of symptom onset [2], thus imposing tremendous economic consequences. According to one study conducted in recent year, the economic burden of RA has increased significantly in such a way that the annual economic burden of RA in the United States was $19.3 billion, where the total annual societal cost would rise to nearly 40 billion after adding the intangible and indirect costs [3].

Not only is RA the cause of a profound economic burden, the concomitant systemic inflammation might be responsible for a wide array of comorbid conditions commonly seen in RA, particularly Sjögren’s Syndrome [1,4,5,6]. One meta-analysis of 19 studies reported the pool prevalence of SS among RA patients was as high as 55% [7]. It is also worth noting that the coexistence of RA and SS may pose a higher disease burden than is typical with RA alone. A prior study noted that RA patients with SS may have higher levels of rheumatoid factor or anti-citrullinated peptide antibody than did those with RA alone [8]. Consequently, one recent nationwide survey disclosed that RA subjects with the concomitant SS may have higher risks of comorbidities including cardiovascular disease, malignancies and serious infections [5]. Thus, there is an urgent need for clinical treatments or drugs with safety and efficacy to lessen the disease progression, particularly the incident SS.

Chinese herbal medicine (CHM), long-used in Asian countries to treat RA, has attracted attention for a long time [9]. With the application of omics and bioinformatics to natural herbs research, it had shown that some herb products could exert inhibitory action on inflammatory cytokines that are involved in the pathogenesis of inflammatory diseases [10,11]. Take the Tripterygium wilfordii hook F (TwHF) for example, former studies showed that combination treatment with TwHF and methotrexate (MTX) was more effective than MTX alone in decreasing the secretions of proinflammatory cytokines and tumor necrosis factor-α (TNF-α) [12,13]. Meanwhile, those receiving a combination of TwHF and MTX experienced better control of disease activity when compared to RA patients treated with MTX alone [14]. In light of the aberrant expression of inflammatory mediators that may link RA and SS [15], understanding the relationship between CHM use and sequent SS risk would be of importance in the solution of treatment and clinical management for RA.

As of now, several studies reported that adding CHM to conventional therapy may be beneficial in improving both symptoms burden and lacrimal and salivary dysfunction caused by SS [16]. In contrast, another recent randomized controlled trial identified a null association between CHM use and relief of SS symptoms [17]. Thus, no consensus has been achieved regarding the effect of CHM on the management of SS. Most of the previous studies were based mainly on self-reported questionnaires and chart reviews, and the number of recruited patients was small [16,17]. In accordance with the belief that “an ounce of prevention is worth a pound of cure,” early exploration of the impact of CHM on reduction of predisposition to SS, particularly RA subjects, should be stressed. Such documentation would provide an empirically robust ground for health care policymakers to initiate strategies to reduce preventable morbidity among people with SS. To this end, we carried out a nested case-control study using a real-world data to answer this question.

## 2. Methods

### 2.1. Data Source and Identification of Study Participants

We applied a retrospective, nested case-control study design using a national health claims from the Longitudinal Health Insurance Database (LHID) in Taiwan. Nowadays, nearly 99% of residents in Taiwan have enrolled in the National Health Insurance Administration Ministry of Health and Welfare’s program [18]. LHID is a data subset of the NHI program and covers the claims of 1 million beneficiaries randomly selected from all beneficiaries under the NHI program. This database contains all NHI enrollment files, claims data, and prescription drug information that provides comprehensive information on all insured subjects.

In the present study, the RA cohort was constructed by identifying all people in the linked claims data set who were 20–80 years of age and had at least three outpatient service claims or one hospital service in which RA was recorded (International Classification of Diseases, 9th Revision, Clinical Modification, ICD-9-CM 714.0) between January 2000 and December 2010. Afterwards, all enrollees were connected to the catastrophic illness registry to ensure diagnostic validity. This is because beneficiaries with major diseases, such as autoimmune disorders, are exempted from the required cost under the NHI program. This approach allowed us to strictly define RA cases and reduce potential misclassification bias. We, therefore, used the date of approval for catastrophic illness registration as the starting point for patients diagnosed with RA. We adhered to the rule that subjects must be excluded if they had any history of SS prior to RA onset. All enrollees were followed up from the date of enrollment to the date of incident SS or the end of follow-up visit, whichever happened first. This article research project was approved by Ethics Committee of the Buddhist Dalin Tzu Chi Hospital (No. B10004021-1) and was conducted with consideration of Helsinki Declaration in all phases of the study. Additionally, the institutional review board waved the need for informed consent for this study since an encrypted database was fully used.

### 2.2. Identification of Case and Control

The primary outcome measure was first-time diagnosis of SS in which they occurred between 2001 and 2013 (ICD-9-CM code 710.2). The documentation of the SS code was regarded as valid if the enrollee has incurred at least twice in the records of outpatient clinics within 1 year or at least one hospitalization during the study period. We also capitalized on the catastrophic illness registry to ensure the accuracy of enrollees’ diagnoses, as conducted in a previous study [19]. We excluded the cases with a diagnosis of SS prior to the onset of RA (*n* = 144), and removed those patients who were followed for less than one year after cohort entry or those who had missing data (*n* = 17). The final cohort comprised 10,710 new-onset RA patients. Of them, each SS case was matched, according to age (within 2 years) and sex, using a risk set sampling of 1:2, with two control subjects who were not diagnosed with SS (Figure 1). The outcome date for each case group was assigned as the index date to the control group, for case and control groups with the same probability to occurrence of SS event during the follow-up period.

### 2.3. Exposure Assessment of CHM Use

To define CHM exposure of subjects, we examined the individual CHM treatment records occurring from the cohort entry date to the index date. Under the NHI program, the medical services used to treat specific diseases that last for 30 days, or more, are considered as one complete course of treatment. In this context, the patients are not required to make copayments for medical services after the first clinic visit throughout the remainder of the treatments. So based on the formerly-established method [20], CHM users were defined as they ever received the relevant CHM treatments being made by the certified Chinese medicine physician for more than 30 days due to RA or its associated symptoms, whereas those visited Western medical doctors only were deemed non-CHM users. For CHM users, we summed up the cumulative days of CHM therapy and categorized them into three levels, low, medium, and high sub-periods, based on the length (in days) of time of receiving CHM therapy, namely, use for 31–365 days, 366–730 days, and 731 days or more. This procedure allowed us to clearly shed light on the dose effect of CHM on the prevention of SS among participants.

### 2.4. Measurement of Covariates

Of the covariates considered for the study, it comprised gender, age, income for estimating insurance payment, urbanization of the subject’s residential area and former comorbidities. Regarding income, we used the premium category as a proxy and it was transformed to ordinal variables, namely New Taiwan Dollars [NTD] ≦ 17,880, 17,881–40,000, and ≥400,001. Furthermore, we adopted the urbanization rate of insured zone studied by former scholars, and ranging from level 1 (highly urban) to level 7 (highly rural), as the standard to assess personal urbanization [21]. Baseline comorbidities for each subject were assessed on the basis of individual medical records that occurred within one year prior to cohort entry, and all of them were evaluated by the established Charlson–Deyo comorbidity index (CCI) [22]. It contains 17 chronic diseases and scores on a score of 1–6, revealing higher total scores indicated severer burdens of comorbidities. Medication use was separated into two subgroups according to if the enrollee had (vs. not used at all) the disease-modifying anti-rheumatic drugs, or corticosteroids, for more than 180 days after RA onset.

### 2.5. Statistical Modeling

For baseline characteristics, continuous variables were represented as the mean (standard deviation, SD) and categorical variables reported as frequencies and percentages. The student’s *t*-test and Chi-square test were used to evaluate whether there was a significant difference between two groups. Univariate conditional logistic regression analysis was used to estimate the crude odds ratio (OR) and the corresponding confidence interval (CI) of SS events among CHM users. Multivariate conditional logistic regression was then performed in which the results were adjusted for all covariates that were measured in one year preceding the index date, which included age, gender, urbanization level, income and comorbidities. Subgroup analysis stratified by sex and age was also performed. All data processing and statistical analyses were performed using SAS version 9.3 for Windows (SAS Institute Inc., Cary, NC, USA). The statistical significance was determined at two-tailed and *p* < 0.05.

## 3. Results

A total of 10,710 RA patients who met the selection criteria during 2000–2010 were identified. Among them, 916 and 1832 matched pairs of RA patients with and without SS were recruited. Baseline characteristics are shown in Table 1. The mean age was 53.2 years (SD = 14.4) and the majority were female (86.9%). Additionally, the majority of enrollees had a monthly income of NTD 17,881–43,900 (52.0%) and lived in urbanized areas (57.2%). Collectively, there were no differences in initial demographic data or comorbidities between two groups.

Of the whole study cohort, after using multivariable logistic regression model to explore the association between previous exposure of CHM use and SS risk by the end of 2013 (Table 2), we observed that those who ever received CHM therapy had a lower risk of SS than those who did not use CHM (adjusted OR = 0.40; 95% CI: 0.34–0.47). Notably, this benefit increased with longer exposure to CHM use, from 56% of those using low intensity CHM, to 58% of those using medium intensity CHM, and to 83% for those receiving high intensity CHM, thus suggesting a dose-dependent inverse relationship between CHM use and SS risk. Table 3 presents these results, stratified by age and sex. Multivariable stratified analysis showed that the benefit of CHM therapy in reducing SS appeared to be more predominant in females, with an adjusted OR of 0.38 (95% CI: 0.32–0.46) (Table 3). Furthermore, of the commonly prescribed CHM formulas, uses of several prescriptions may be related to the lower risk of SS, which contained Da Huang, Shu-Jing-Huo-Xue-Tang (SJHXT), Du-Huo-Ji-Sheng-Tang (DHJST), Ge-Gen-Tang (GGT), Ping-Wei-San (PWS), Shao-Yao-Gan-Cao-Tang (SYGCT), and Zhi-Gan-Cao-Tang (ZHCT) (Figure 2).

## 4. Discussion

SS is a chronic autoimmune disease characterized by autoantibody production and lymphocytic infiltration, which has been well recognized as one important extra-articular feature of RA to exacerbate the negative clinical prognosis for RA patients. Faced with few specific strategies of prevention of SS in the standard treatment, exploring alternative treatments is of great therapeutic interest. As a whole, in this over 10-year follow-up study, we had provided the first evidence to indicate the integration CHM into the standard treatments was related to the lower risk of having SS. Notably, this benefit could be increased with a long-term exposure to CHM use, from 56% for those using CHM for 31–365 days, to 58% for those using CHM for 366–730 days, and to 83% for those receiving CHM for more than two years. The establishment of dose-response relationship in this observational study may support the causal association between exposure and disease. Despite the lack of comparable literature, the positive effect of CHM on SS prevention among these patients could add to the growing body of literature on the clinical efficacy of complementary therapies among patients diagnosed with rheumatic diseases [23,24,25]. A variety of natural products from traditional Chinese medicine have been shown to possess effective anti-inflammatory along with antiarthritic activities [26,27], which may explain the beneficial effect of CHM found in our work.

Findings of the present study indicated that female patients benefited more from CHM use than did males. As others have shown, females often display better knowledge, attitudes, and self-care practices [28], and accordingly, they may tend to adhere to the prescribed medical regimen, thus decreasing their chance of developing SS. In addition, sex hormones, especially estrogen, have been shown to exert anti-inflammatory effects. One study showed that high levels of estrogens were beneficial in downregulating the expression of the inflammatory mediators [29], which has been proven to take on the role of development of SS [11].

Of the commonly used single-herb products to treat RA, we noted that the prescription of Da-Huang might lessen the risk of SS. Pharmacological studies have shown that Da-Huang dose-dependently moderates the release of nitric oxide in lipopolysaccharide-stimulated macrophage RAW264.7 cells and remarkably reduces IL-6 and IL-1β secretion via mediation of the PI3K-Akt signaling pathway [30]. This action could account for the positive effect of Da-Huang observed in this study. Of the commonly used multi-herb products, we observed that DHJST use was associated with a decreased chance of SS among RA patients. A previous study reported that DHJST exerts a powerful anti-inflammatory effect [31]. The relevant mechanisms by which DHJST inhibits expression of cytokines, like TNF-α, IL-1β and IL-6, may involve regulation of the toll-like receptor 4 (TLR4)/NF-κB signaling pathway [32]. The TLR4/NF-κB signaling pathway plays a central role in driving inflammation via the differentiation and amplification of T helper17 together with over-expression of inflammatory mediators [33,34], thus promoting susceptibility to SS.

The current study pointed to a lower incidence of SS among RA patients who used SJHXT and SYGCT. Clinically, these CHM products are often prescribed to arthritis patients for the treatment of muscle pain. One earlier study in a rodent model suggested that SJHXT may intensify anti-inflammatory and analgesic effects by modulating the activity of the α-2 adrenoceptor [35]. A review article reported that dysregulation of the α2- adrenoceptor pathway may contribute to the aberrant cytokine gene expression [36]. SYGCT use also correlated with a lower risk of SS. Chang and colleagues reported that this compound markedly inhibited the production of inflammatory mediators in rats with polycystic ovary syndrome by blocking TLR4/NF-*κ*B signal pathway [37]. This pathway promotes proinflammatory activities in immune cells, thereby leading to a variety of other inflammatory and autoimmune disorders [34].

An association between ZGCT and PWS use and a decreased rate of SS development in RA patients was reported as well. Several previous animal experiments have shown that the anti-inflammatory properties of these compounds were also present in compound extracts, including *Radix glycyrrhiza* from ZGCT [38], and *Magnolia officinalis* from PWS [39]. The mechanisms by which these ingredients markedly decrease the secretion of inflammatory cytokines may be due in part to the inhibition of NF-κB and mitogen-associated kinase signaling pathways [38,39]. Ge-Gen-Tang is proposed to exert anti-oxidant and anti-inflammatory activities by suppressing inflammatory signaling. Puerarin, a major ingredient of GGT, has proven to suppress inflammatory mediator release by blocking NF-*κ*B in lipopolysaccharide-induced peripheral blood mononuclear cells [40], decreasing to some extent the risk of SS.

Despite its obvious strengths, several important limitations restrict the significance of our study. First, information regarding family history, lifestyle, body weight, exercise and laboratory parameters were not recorded in the database. Thus, it is possible that one or more confounding variables may be partly responsible for this association. Therefore, caution should be exerted when interpreting the findings, especially regarding daily drug dosage. A randomized controlled trial is warranted in order to examine, more thoroughly, the potential mechanisms underlying the clinical benefits of CHM products in controlling the development of SS. Second, the findings herein are merely based on a nested case-control design within a retrospective cohort study that uses ICD-9-CM diagnostic codes. Thus, bias due to miscoding and misclassification may arise. To minimize this potential error, we selected subjects with either RA or SS only after they were recorded as having at least three ambulatory or inpatient claims reporting consistent diagnoses. It should also be acknowledged that the NHI Bureau of Taiwan has randomly reviewed the charts and audited all medical charges, and given heavy penalties for outlier charges or malpractice to validate the quality of data. Additionally, as the probability of individuals being misclassified is equal for the two groups, a non-differential misclassification would only result in bias toward the null-value. Third, data regarding RA severity were unavailable in the database, and failure to examine this factor might bias any conclusions. To address this concern, we utilized a proxy indicator to confirm RA severity. The indicator was comprised of the prescriptions of biological agents which included adalimumab, etanercept, infliximab, rituximab and tocilizumab. Findings from the reanalysis showed that those who did use CHM still had a lower risk of SS than those who were not receiving CHM treatment (adjusted OR = 0.45; 95% CI = 0.31–0.58), which implied that the severity of RA dose not alter the direction of association between CHM treatment and likelihood of SS. After acknowledging the limitations of the research, this study has several strengths that bolster the value of the findings. The first strength stems from the use of a large population database. Over 90% of the Taiwanese population and healthcare providers are covered by the NHI program, which includes a representative Taiwanese RA sample, leaving little room for non-response or loss to follow-up, especially given the relatively low incidence of RA in the population. The second merit stems from the long observation time used in our study. SS is a chronic disease and the employed longer than10-year follow-up period allowed us ample opportunity to observe and assess the underlying correlation. Lastly, the nested case-control approach used is a rival alternative to a cohort analysis when studying time-dependent exposure, as in the use of CHM treatments. Hence, our study reflects real-world data that approximate those present in clinical reports using a randomized controlled trial.

## 5. Conclusions

To summarize our findings, this population-based nested case-control study revealed that the integration of CHM, during routine treatment of RA, lessens the risk of developing SS by approximately 60%. We believe that these findings may help to plan interventions to make complementary therapies more responsive to the needs of individuals living with RA. Future research efforts should adopt prospective randomized trials to overcome the disadvantages of this study, thus providing more robust insights in clinical practice.

## Figures and Tables

**Figure 1 medicina-59-00683-f001:**
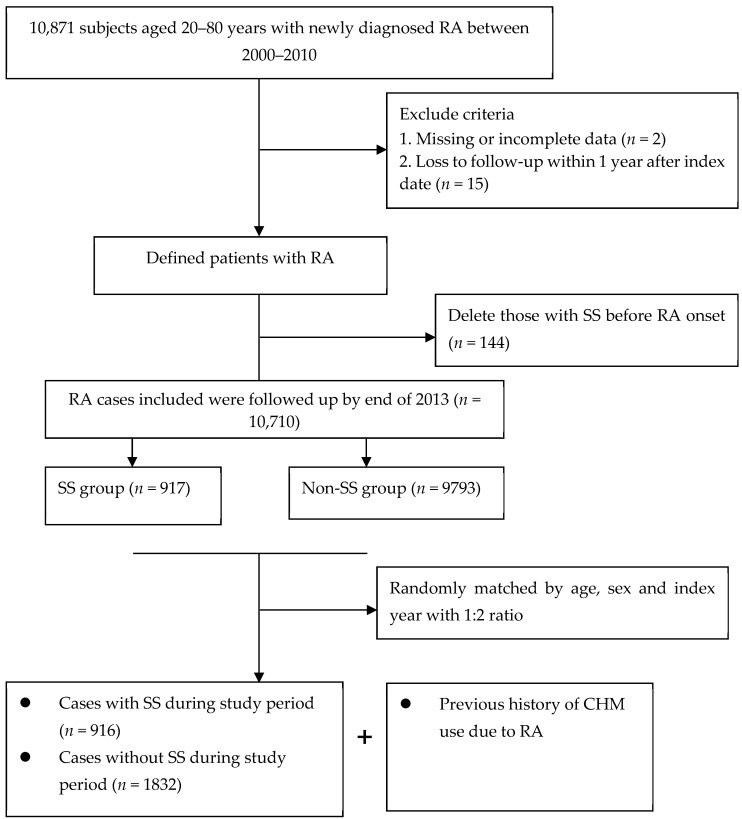
Flowchart of patient screening.

**Figure 2 medicina-59-00683-f002:**
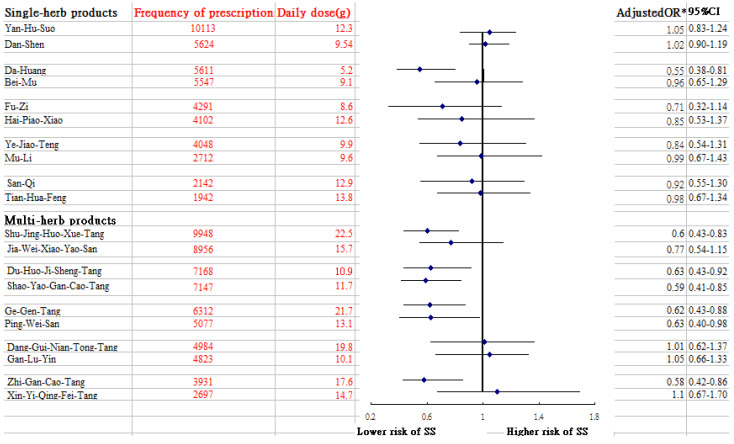
Risk of SS in relation to the 10 most-used single-herb and multi-herb CHM products for RA patients. * Model adjusted for age, residential area, monthly income, medication use and CCI.

**Table 1 medicina-59-00683-t001:** Demographic data and selected comorbidities between two groups.

Variables	Total Group	Case	Control	*p*
		N = 916 (%)	N = 1832 (%)	
Age (years)				0.47
≤50	1254 (45.6)	427 (46.6)	827 (45.1)	
>50	1494 (54.4)	489 (53.4)	1005 (54.9)	
Mean (SD)	53.2 (14.4)	53.1 (14.8)	53.3 (13.9)	0.81
Sex				0.99
Female	2388 (86.9)	796 (86.9)	1592 (87.0)	
Male	360 (13.1)	120 (13.1)	240 (13.0)	
Monthly income				0.53
Low	1210 (44.1)	406 (44.3)	804 (43.9)	
Median	1430 (52.0)	469 (51.2)	961 (52.5)	
High	108 (3.9)	41 (4.5)	67 (3.7)	
Residential area				0.66
Urban	1572 (57.2)	516 (56.2)	1057 (57.7)	
Suburban	416 (15.1)	128 (15.9)	270 (14.7)	
Rural	760 (27.7)	272 (27.9)	505 (27.6)	
Medication use				0.67
Yes	2038 (74.2)	684 (74.7)	1354 (73.9)	
No	710 (25.8)	232 (25.3)	478 (26.1)	
CCI	4.73 (7.9)	4.59 (7.8)	4.86 (8.0)	0.40

CCI: Charlson–Deyo Comorbidity Index; SD: standard deviation.

**Table 2 medicina-59-00683-t002:** The association between SS onset and use of CHM.

CHM Exposure	Subjects				Crude OR (95% CI)	Adjusted OR * (95% CI)
	Case *n* = 916		Control *n* = 1832			
Non-CHM users	659	72%	945	52%	1	1
CHM users	257	28%	887	48%	0.42 (0.35–0.49)	0.40 (0.34–0.47)
Low intensity (31–365 days)	215	23%	720	39%	0.44 (0.30–0.64)	0.44 (0.30–0.65)
Medium intensity (366–730 days)	35	4%	112	6%	0.43 (0.36–0.51)	0.42 (0.35–0.50)
High intensity (731 days or more)	7	1%	55	3%	0.18 (0.09–0.34)	0.17 (0.10–0.32)

OR: odds ratio; CI: confidence interval; CHM: Chinese herbal medicine. * Model adjusted for age, residential area, monthly income, medication use and CCI.

**Table 3 medicina-59-00683-t003:** SS risk for RA patients with and without CHM use stratified by sex and age.

Variables	Subjects, *n* (%)	Crude OR (95% CI)	Adjusted OR * (95% CI)
Female			
Non-CHM users	566 (71.1)	1	1
CHM users	230 (28.9)	0.39 (0.33–0.47)	0.38 (0.32–0.46)
Male			
Non-CHM users	93 (77.5)	1	1
CHM users	27 (22.5)	0.62 (0.37–1.02)	0.63 (0.37–1.03)
Age ≤ 50			
Non-CHM users	291 (68.1)	1	1
CHM users	136 (31.9)	0.38 (0.30–0.48)	0.37 (0.29–0.47)
Age > 50			
Non-CHM users	368 (75.3)	1	1
CHM users	121 (24.7)	0.44 (0.35–0.56)	0.42 (0.35–0.49)

OR: odds ratio; CI: confidence interval; CHM: Chinese herbal medicine. * Model adjusted for age, residential area, monthly income, medication use and CCI.

## Data Availability

The datasets analyzed in this article are not publicly available. Data are available from the National Health Insurance Research Database (NHIRD) published by Taiwan National Health Insurance (NHI) Bureau. Due to legal restrictions imposed by the government of Taiwan in relation to the “Personal Information Protection Act”, data cannot be made publicly available. Requests to access the datasets should be directed to the NHIRD and the corresponding authors.

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
