# Peer review of "Relationship between Chinese Herbal Medicine Use and Risk of Sjögren’s Syndrome in Patients with Rheumatoid Arthritis: A Retrospective, Population-Based, Nested Case-Control Study"

_medicina, 2023, doi:10.3390/medicina59040683_

Round 1

Reviewer 1 Report

Thanks for receiving the opportunity to review your paper. It deals with an interesting and relevant topic.

In introduction expanded reflection on the significance of the study and novelty of the study in respect to previous researches should be included.

In methodology, mention the total duration of the study conducted. Also, clarify regarding whether the subjects were allowed to take treatment/medicine for their comorbid conditions parallel to CHM.

In the result section, abbreviation should be added below the tables and explanation for figure 2 is not much clear. Also, state about the age group who showed comparatively more significant result.

In the discussion, it is mentioned that the improvement was 80% but in conclusion, it is mentioned 60%. Explain.

I suggest correcting English language revisions and other typos and grammar errors.

Author Response

Dear Editor-in-Chief

Thank you for your positive response and the constructive comments from the Reviewers. We followed closely the Reviewers’ comments and the Journal's guidelines for authors in revising our manuscript. In the revised manuscript, we highlighted all amendatory material by red color in MS Word. Our point-by-point responses to the Reviewers’ concerns and suggestions are listed below.

Q1. In introduction expanded reflection on the significance of the study and novelty of the study in respect to previous researches should be included.

Response: As suggested by the Reviewer, we rewrote the points pertaining to the merit of study. Please refer to Lines 7-17 on Page 6.

Q2. In methodology, mention the total duration of the study conducted. Also, clarify regarding whether the subjects were allowed to take treatment/medicine for their comorbid conditions parallel to CHM.

Response: We added information to address this issue and avoid possible confusion. Please refer to the 25th line of Page 9. On the other hand, as shown in Table 1, we found that there was no significant difference between two groups in age, sex, monthly income, residential area, medication usage and number of comorbidities after applying the matching procedure, indicating that the two groups were comparable on these characteristics. Accordingly, the prior prescriptions used to manage individual comorbid conditions suggested that there was no prejudice in the findings of this study. Nonetheless, in order to address this concern, we recruited RA subjects without any comorbidities to reanalyze the relationship of interest. The re-analysis indicated that, as compared to RA patients without CHM use, the selected group of RA patients with CHM use still had a significantly lower OR for SS (adjusted OR=0.43; 95%CI: 0.36-0.51). Despite our careful efforts to control for confounding factors in this work, we still agree that the evidence from any observational cohort study is generally less robust than that from randomized trials because cohort study designs could be subject to unpredictable biases that may arise from unmeasured or unknown confounders. These points were noted in the Discussion part. Please refer to the sentences extending from Line 24 of Page 12 to the 4th line of Page 13.

Q3. In the result section, abbreviation should be added below the tables and explanation for figure 2 is not much clear. Also, state about the age group who showed comparatively more significant result.

Response: After examination by statistical approach, the beneficial effect of CHM in reducing onset of SS still remained, regardless of age group. In addition, we added the full name to account for the abbreviations shown on the whole tables, and we have improved the image resolution of Figure 2. Please refer from Page 21 to Page 23 along with Page 25.

Q4. In the discussion, it is mentioned that the improvement was 80% but in conclusion, it is mentioned 60%. Explain.

Response: We appreciate the Reviewer's comment. In this work, the overall effect of CHM in reducing onset of SS showed an adjusted odds ratio of 0.40 (95% confidence interval: 0.34-0.47). Notably, this benefit could be intensified with long-term exposure of CHM use, from 56% of those using CHM with low intensity, to 58% of those using CHM with medium intensity, and to 83% for those receiving CHM with high intensity. To avoid any unnecessary confusion, we carefully rewrote  this point. Please refer to Lines 8-11 on Page 10.  

Q5. I suggest correcting English language revisions and other typos and grammar errors.

Response: We appreciate the Reviewer's valuable comments and made every effort to address this concern. Additionally, before submitting the revision, this paper has been carefully edited for spelling and grammar by a professional editor who is a native English speaker and a member of the American Medical Writers Association, as well as the author of nearly 200 English-language refereed articles.

Author Response

Dear Editor-in-Chief

Thank you for your positive response and the constructive comments from the Reviewers. We followed closely the Reviewers’ comments and the Journal's guidelines for authors in revising our manuscript. In the revised manuscript, we highlighted all amendatory material by red color in MS Word. Our point-by-point responses to the Reviewers’ concerns and suggestions are listed below.

Q1. 32…syndrome [1] is….it is not appropriate to add a reference in the abstract

Response: The concern mentioned by the Reviewer have been carefully amended. Please refer to Line 2 on Page 3 in the revised version.

Q2. 36.. SS for...the meaning of the abbreviation should be mentioned first and then the abbreviation used later on.

Response: We added an abbreviation of the Sjögren’s syndrome and showed this on the first line of the ABSTRACT. Please refer to Line 2 on Page 3.

Q3. 80.. Also worth nothing...English editing and grammar revision are needed in the

whole manuscript 

Response: The sentence indicated by the Reviewer has been amended. Please refer to Lines 14-15 on Page 5. Before submitting the revision, this paper was carefully edited for spelling and grammar by a professional editor who is a native English speaker and a member of the American Medical Writers Association, as well as the author of nearly 200 English-language refereed articles.

Q4. 105.. most of previous studies...add the references of these studies. The limitations of the study should be mentioned. The recommendations of the study should be added 

Response: We added more references to expand on our earlier discussion. Please refer to Lines 10-12 on Page 6. Additionally, the Limitations part has been highlighted in the revised manuscript. Please refer to the section extending from the 24th line of Page 12 to the 19th line on Page 13.  

Reviewer 3 Report

Thank you for the opportunity to review your manuscript.

ABSTRACT

Please clarify what SS stands for.

I suggest better clarifying in the abstract CHM dose and frequency of therapy. 

INTRODUCTION

Please summarize the first paragraph of the introduction and revise the grammar.

Line 103 please rephrase.

Line 106 please check the grammar.

Line 111-112 please proofread.

MATERIALS AND METHODS

Line 125 please check the grammar.

Line 129 I suggest better explaining the sentence's content.

Line 138 please check the grammar.

Figure 1. Please draw again in a clearer way.

RESULTS

Figure 2 is not in high quality, please replace it.

DISCUSSION

In the discussion section, I suggest introducing a paragraph that discusses other traditional and integrative medicines for the treatment of RA (for example, I suggest citing: Masiero, S., Maccarone, M.C. Health resort therapy interventions in the COVID-19 pandemic era: what next?. Int J Biometeorol 65, 1995–1997 (2021). https://doi.org/10.1007/s00484-021-02134-9; Tognolo, L.; Coraci, D.; Fioravanti, A.; Tenti, S.; Scanu, A.; Magro, G.; Maccarone, M.C.; Masiero, S. Clinical Impact of Balneotherapy and Therapeutic Exercise in Rheumatic Diseases: A Lexical Analysis and Scoping Review. Appl. Sci.2022,12,7379. https://doi.org/ 10.3390/app12157379; Maccarone, M.C., Magro, G., Solimene, U. et al. From in vitro research to real life studies: an extensive narrative review of the effects of balneotherapy on human immune response. Sport Sci Health 17, 817–835 (2021). https://doi.org/10.1007/s11332-021-00778-z). Additionally, I recommend better explaining the role of dose-response in these patients, beyond the time for which the CHM has been taken. Otherwise, it may be difficult for inexperienced readers to apply what they have learned from reading the paper in a clinical setting. It is suggested to integrate CHM in RA patients to prevent SS onset: for how long? At what doses? Assuming what in particular? I advise adding this information to the discussion. Furthermore, the optimal dose, duration, and combination of CHM with conventional therapies remain unclear and may vary among patients. Please discuss this point.

CONCLUSION

Please rewrite the conclusions, summarizing in more detail the message contained in the paper.

Author Response

Dear Editor-in-Chief

Thank you for your positive response and the constructive comments from the Reviewers. We followed closely the Reviewers’ comments and the Journal's guidelines for authors in revising our manuscript. In the revised manuscript, we highlighted all amendatory material by red color in MS Word. Our point-by-point responses to the Reviewers’ concerns and suggestions are listed below.

Reviewer 3

ABSTRACT

Q1. Please clarify what SS stands for. I suggest better clarifying in the abstract CHM dose and frequency of therapy.

Response: We added an abbreviation of the Sjögren’s syndrome and showed this on the first line of the ABSTRACT. Please refer to the second line on Page 3. Due to the limit on the number of words in the ABSTRACT, the CHM dose and frequency of therapy that the Reviewer suggested have been showed in Figure 2 rather than in the ABSTRACT. Please refer to Page 25.

INTRODUCTION

Q2. Please summarize the first paragraph of the introduction and revise the grammar.

Line 103 please rephrase.

Line 106 please check the grammar.

Line 111-112 please proofread.

Response: We amended this content as suggested by the Reviewer. Please refer to Lines 7-17 of Page 6.  

MATERIALS AND METHODS
Q3.
Line 125 please check the grammar.

Response: This omission has been addressed. Please refer to Lines 3-5 on Page 7.

Q4. Line 129 I suggest better explaining the sentence's content.

Response: We clarified the sentences indicated by the Reviewer. Please refer to Lines 7-11 on Page 7.

Q5. Line 138 please check the grammar.

Response: We are grateful for the Reviewer's suggestion and e rewrite this sentence. Please refer to the 14th line of Page 7.

Q6. Figure 1. Please draw again in a clearer way.

Response: After careful group discussion, we decided that the current version of Figure 1 is more beneficial in indicating the nature of nested case-control study used herein.

RESULTS
Q7. Figure 2 is not in high quality, please replace it.

Response: We improved the image resolution of Figure 2 based on the Reviewer's comment. Please refer to Page 25 in the revised paper.

DISCUSSION
Q8. In the discussion section, I suggest introducing a paragraph that discusses other traditional and integrative medicines for the treatment of RA (for example, I suggest citing: Masiero, S., Maccarone, M.C. Health resort therapy interventions in the COVID-19 pandemic era: what next?. Int J Biometeorol 65, 1995–1997 (2021). https://doi.org/10.1007/s00484-021-02134-9; Tognolo, L.; Coraci, D.; Fioravanti, A.; Tenti, S.; Scanu, A.; Magro, G.; Maccarone, M.C.; Masiero, S. Clinical Impact of Balneotherapy and Therapeutic Exercise in Rheumatic Diseases: A Lexical Analysis and Scoping Review. Appl. Sci.2022,12,7379. https://doi.org/ 10.3390/app12157379; Maccarone, M.C., Magro, G., Solimene, U. et al. From in vitro research to real life studies: an extensive narrative review of the effects of balneotherapy on human immune response. Sport Sci Health 17, 817–835 (2021). https://doi.org/10.1007/s11332-021-00778-z). Additionally, I recommend better explaining the role of dose-response in these patients, beyond the time for which the CHM has been taken. Otherwise, it may be difficult for inexperienced readers to apply what they have learned from reading the paper in a clinical setting. It is suggested to integrate CHM in RA patients to prevent SS onset: for how long? At what doses? Assuming what in particular? I advise adding this information to the discussion. Furthermore, the optimal dose, duration, and combination of CHM with conventional therapies remain unclear and may vary among patients. Please discuss this point.

Response: As suggested by the Reviewer, we incorporated these references into the main text. The detailed explanation of the dose-response relationship has been described as well. Please refer to Lines 1-7 of Page 11, together with the updated references lists.

  Also, the relevant dose and frequency of herbal formulae have been displayed in Figure 2. Meanwhile, we highlighted that the dose of CHM used per day represents the general use pattern of herbal products in treating RA by certified Taiwanese Chinese medicine physicians, and not directly indicating these doses found effective in preventing SS. We anticipate that this pilot study could be used as a basis for further pharmacological investigations and clinical trials. We rewrote part of the discussion to clarify our findings. Please refer to Lines 1-4 of Page 13. 

CONCLUSION
Q9.
Please rewrite the conclusions, summarizing in more detail the message contained in the paper.

Response: As suggested by the Reviewer, we rewrote the CONCLUSION section to more accurately describe the merit of our study. Please refer to Lines 7-12 on Page 14.

Round 2

Reviewer 2 Report

The manuscript improved to a great extent 

Reviewer 3 Report

The manuscript can be published.